# Parametric and Nonparametric PI Controller Tuning Method for Integrating Processes Based on Magnitude Optimum

**Tomaž Kos** [1,2,*], **Mikuláš Huba** [3] and **Damir Vrančić** [1]

[1] Department of Systems and Control, Jožef Stefan Institute (JSI), Jamova cesta 39, 1000 Ljubljana, Slovenia; damir.vrancic@ijs.si

[2] Jožef Stefan International Postgraduate School, Jamova cesta 39, 1000 Ljubljana, Slovenia

[3] Department of E-Mobility, Automation and Drives, Institute of Automotive Mechatronics, Faculty of Electrical Engineering and Information Technology, Slovak University of Technology (STU), Ilkovičova 3, 812 19 Bratislava, Slovakia; mikulas.huba@stuba.sk

[*] Correspondence: tomaz.kos@ijs.si

**Abstract:** Integrating systems are frequently encountered in the oil industry (oil–water–gas separators, distillation columns), power plants, paper-production plants, polymerisation processes, and in storage tanks. Due to the non-self-regulating character of the processes, any disturbance can cause a drift of the process output signal. Therefore, efficient closed-loop control of such processes is required. There are many PI and PID controller tuning methods for integrating processes. However, it is hard to find one requiring only a simple tuning procedure on the process, while the tuning method is based either on time-domain measurements or on a process transfer function of arbitrary order, which are the advantages of the magnitude optimum multiple integration (MOMI) tuning method. In this paper, we propose the extension of the MOMI tuning method to integrating processes. Besides the mentioned advantages, the extension provides efficient closed-loop control, while PI controller parameters calculation is still based on simple algebraic expressions, making it suitable for less-demanding hardware, like simpler programmable logic controllers (PLC). Additionally, the proposed method incorporates reference weighting factor *b* that allows users to emphasize either the disturbance-rejection or reference-following response. The proposed extension of the MOMI method (time-domain approach) was also tested on a charge-amplifier drift-compensation system, a laboratory hydraulic plant, on an industrial autoclave, and on a solid-oxide fuel-cell temperature control. All closed-loop responses were relatively stable and fast, all in accordance with the magnitude optimum criteria.

**Keywords:** magnitude optimum; PID control; PI controller; controller tuning; integrating processes; 2-DOF

## 1. Introduction

The most used control algorithm in the industry is PID control. It is widely used in various processes due to its tuning simplicity, good control performance, and robustness in a wide range of operating conditions. Disturbance-rejection and set-point tracking performance depends on controller tuning. In addition to typical tuning rules, such as Cohen–Coon, Chien–Hrones–Reswick, Ziegler–Nichols, or refined Ziegler–Nichols rules, more advanced methods have also been proposed. Usually, they use more complex algorithms to identify the processes [1–6].

Integrating systems contain at least one pole in the origin. Since they are not self-regulating, in the case of process input disturbance, process output might drift. Consequently, efficient control

of integrating processes (IP) is a challenging task. There are different types of integrating systems that can be classified according to the number of poles in the origin and location of other poles in the transfer function. Integrating systems are frequently encountered in the industry. For example, representatives of industrial processes that exhibit pure integrator plus time-delay (ITD) transfer function models are oil–water–gas separators in the oil industry [7], bottom-level control in a distillation column [8], composition control loop of a high-purity distillation column [9], heat-integrated distillation columns [10], isothermal continuous copolymerisation reactors [11], storage tanks with an outlet pump [12], pulp and paper plants [7], and high-pressure steam flowing to a steam-turbine generator in a power plant [13]. First-order integrating systems with/without delay (IFOTD) can be encountered in liquid-storage tanks [14], a jacketed continuous stirred tank reactor (CSTR) carrying out an exothermic reaction [15], and paper-drum-dryer cans [16].

Several tuning methods for IP have been developed so far [17,18]. They can be classified according to models of integrating systems, i.e., tuning methods for integrating systems with time-delay [7,19–28], for first-order integrating systems with time-delay [18,29–51], for higher-order integrating systems with time-delay [52], and nonparametric tuning methods for integrating systems [53–56].

Tunings methods for integrating systems with time-delay can also be applied to a first- or higher-order integrating system by using a relay feedback identification method to model the system as an integrating system with time-delay [21,24,26,27]. The method of Mercader and Baños [19] takes into account process uncertainties. The PD controller, based on the Smith predictor scheme with gain- and phase-margin specifications for an integrating-with-time-delay (ITD) process, was proposed by Chakraborty et al. [21]. Raza et al. [26] developed a tuning rule derived in terms of maximal sensitivity ($M_S$) that was based on pole-placement and frequency-response matching criteria. Visioli [7,20] proposed a three-state and PID controller structure to obtain a time-optimal set-point response. The process model is neatly calculated by the least-squares-based identification technique during a process set-point change. Therefore, the process model does not have to be known a priori, before changing the process set point.

Most tunings methods for first-order integrating systems with time-delay can also be applied in higher-order systems by approximating the process as an IFOTD model [30,32–35,39,40,43–45,47,48,50, 51]. Controller design based on IMC PID tuning rules was proposed by Kumar and Padma Sree [18], Skogestad [39], Jin and Liu [43], Panda [41], Ghousiya Begum et al. [35], and Najafizadegan et al. [34]. The IMC filter time constant in the Kumar and Padma Sree [18] method was chosen to set the trade-off between performance and robustness of tracking and control responses. Similarly, it was used to choose the robustness degree in the Jin and Liu [43] method. Anil and Padma Sree [44] developed a PID controller tuning method for IFOTD processes with or without additional process zero. The method is based on a pole-placement strategy, while the tuning parameter is the maximal sensitivity ($M_S$) value. This is similar for the Medarametla and Komanapalli [32] method. Srivastava and Pandit [31] developed a PID controller with a unique set-point filter that created a two-degrees-of-freedom (2-DOF) control system. Controller tuning is based on the linear quadratic regulator (LQR) using a dominant pole-placement approach to obtain a good regulatory response. Bingul and Karahan [49] tuned a PID controller using particle-swarm-optimisation (PSO) and artificial-bee-colony (ABC) algorithms. Atic et al. [51] proposed a tuning method where the controller parameters are determined by obtained stability-boundary loci. Controller parameters for IFOTD or unstable first- and second-order processes with time-delay in a discrete domain were developed by Wang et al. [42]. A predictor-based 2-DOF control design was proposed by Wang et al. [36]. The proposed control-system design was based on using a dead-time compensator (DTC) to predict non-minimal-phase (NMP) dynamics. Some of the tuning methods set the controller integrating gain to zero (Eriksson et al. [38], Kuzishchin et al. [46]). Those methods are not suitable for rejecting process input disturbances. A modified series cascade control structure (SCCS) with two PI and one P controller for a class of IP models with/without positive zero was proposed by Raja and Ali [29]. PI and P controller parameters are obtained using the method of moments and Routh–Hurwitz stability criterion, respectively.

Papadopoulos [52] proposed the tuning of PID controllers for general IP by using the symmetrical-optimum method. The method is effective for disturbance rejection, while it results in higher process output overshoots on reference changes.

Nonparametric controller-tuning methods are model-free (data-based), i.e., they directly exploit a set of closed- or open-loop-process data without requiring the use of a process model. Jeng [53] proposed a PI/PID controller method that was based on the direct synthesis approach and specification of the desired closed-loop transfer function for disturbances. The set-point response can be independently improved by using set-point weighting in the PID controllers. Dey, Mudi, and Simhachalam [54] developed an autotuning PD controller that adjusts proportional and derivative gains to achieve improved overall performance during set-point changes and load disturbances. Due to the lack of an integrator, the tuning rule is not suitable for rejecting process input disturbances. Mataušek and Šekara [56] proposed PI and PID tuning rules on the basis of an extended Ziegler–Nichols experiment (besides ultimate gain and frequency, the steady-state gain and angle of the tangent to the process Nyquist curve at ultimate frequency should be obtained). Controller parameters are calculated by means of optimisation (solution of two nonlinear algebraic equations).

Most of the mentioned methods require the exact process model, they are limited to a few IP models, or the calculation of controller parameters requires optimisation.

One of the more sophisticated tuning approaches is the magnitude optimum (MO) method [57–59]. The MO method can be applied to various processes frequently encountered in chemical and processing industries. Tracking and disturbance-rejection closed-loop responses are usually fast without oscillations [60]. The applicability of the MO method has been extended by using a nonparametric approach in the time-domain instead of using explicit parametric identification of the process. The extension is based on multiple integrations of process input and output signals, and is hence called the magnitude optimum multiple integration (MOMI) method [61]. The MOMI method extracts the required information from a simple time-domain experiment while retaining all advantages of the MO method [60].

Since the original MO method was developed for stable (non-integrating) processes, the MOMI method cannot be used for IP. Namely, when using a one-degree-of-freedom (1-DOF) controller structure [62], MO criteria result in integrating gain of the PI(D) controller equal to zero. Namely, proportional (P) and PD controllers are not capable of rejecting input disturbances. However, in the paper, we show that MO criteria can be met by using 2-DOF PI controllers, similar to the case when optimising disturbance-rejection performance [60]. A 2-DOF controller implements reference weighting factor *b*.

The advantages of an extension of the MOMI tuning method to integrating processes are a tuning formula in closed form for an IP of arbitrary order with delay, and the selection of the process data in the time-domain (given by the process open- or closed-loop time responses) or in the frequency domain (given by the arbitrary-order process transfer function). Moreover, the frequency- and time-domain approaches are equivalent, and the latter does not introduce any errors in the calculation of controller parameters (according to MO criteria). Reference weighting factor *b* allows users to emphasize either disturbance-rejection or reference following in a continuous matter. Besides the mentioned advantages, the extension provides efficient closed-loop control, while PI controller parameters calculation is still based on simple algebraic expressions, making it suitable for less demanding hardware, like slower PLC controllers.

The content of this paper is organised as follows. First, the MOMI tuning method for IP is presented. In Section 3, the stability and robustness of the MOMI tuning method are evaluated. Section 4 provides examples on several different process models and compares the proposed method to other tuning methods made for IP. Real-time experiments in the time-domain on a charge-amplifier drift-compensation system, a laboratory plant, on an industrial autoclave, and on a solid-oxide fuel-cell temperature control are outlined in Section 5.

## 2. MOMI Tuning Method for Integrating Processes

The process in a closed-loop configuration with a 2-DOF controller is presented in Figure 1. Signals $d$, $y$, $r$, and $u$ represent input disturbance, process output, controller reference, and controller output, respectively.

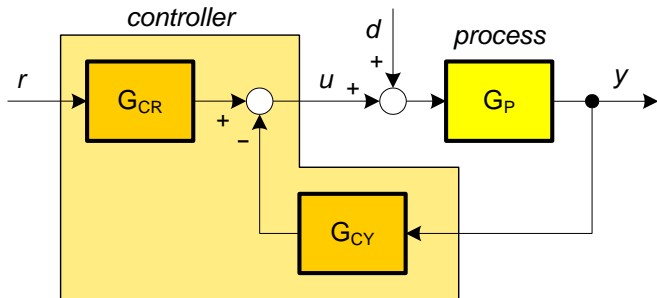

**Figure 1.** General closed-loop scheme with a two-degrees-of-freedom (2-DOF) controller.

The stable process was modelled with the rational transfer function:

$$G_P(s) = \frac{K_{PR}}{s} \frac{1 + b_1 s + b_2 s^2 + \cdots + b_m s^m}{1 + a_1 s + a_2 s^2 + \cdots + a_n s^n} e^{-sT_{delay}},$$ (1)

where $T_{delay}$ represents the time-delay. According to [58], "one possible design aim is to maintain the closed-loop magnitude response curve as flat and as close to unity for as large a bandwidth as possible" (see Figure 2).

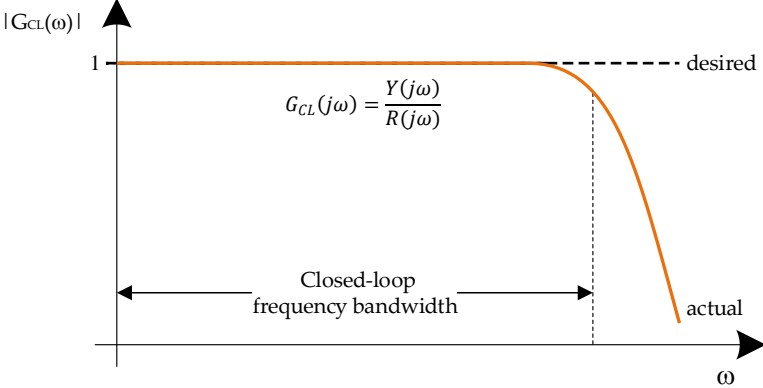

**Figure 2.** "Closed-loop amplitude (magnitude) response over frequency". Adapted from [58].

This technique is called magnitude optimum (MO), modulus optimum, or Betragsoptimum, and results in a fast and non-oscillatory closed-loop time response for a large class of process models [60].

If we have closed-loop transfer function

$$G_{CL}(s) = \frac{Y(s)}{R(s)} = \frac{G_P(s)G_{CR}(s)}{1 + G_P(s)G_{CY}(s)},$$ (2)

the controller is determined in such a way that

$$G_{CL}(0) = 1$$ (3)

$$\lim_{\omega \to 0} \left[ \frac{d^{2k} |G_{CL}(j\omega)|^2}{d\omega^{2k}} \right] = 0; k = 1, 2, \cdots, k\_{max} \tag{4}$$

for as many $k$ as possible [58,61]. The fulfilment of Equation (3) is simple. When using the controller structure containing an integral term, the steady-state control error becomes zero (under the condition that the closed-loop response is stable). Controller order (number of controller parameters) impacts the number of satisfied conditions in Equation (4).

If the closed-loop transfer function is described by equation

$$G_{CL}(s) = \frac{f_0 + f_1 s + f_2 s^2 + \cdots}{e_0 + e_1 s + e_2 s^2 + \cdots}, \tag{5}$$

then Expressions (4) can be met by satisfying the following conditions [60]:

$$\sum_{i=0}^{2n} (-1)^{i+n} \left( f_i f_{2n-i} e_0^2 - e_i e_{2n-i} f_0^2 \right) = 0; n = 1, 2, \ldots \tag{6}$$

The next step is the calculation of the 2-DOF PI controller parameters. A controller may be defined by the next transfer functions:

$$\begin{aligned} G_{CR}(s) &= bK_P + \frac{K_i}{s} \\ G_{CY}(s) &= K_P + \frac{K_i}{s}, \end{aligned} \tag{7}$$

where $b$, $K_P$, and $K_i$ are the reference weighting factor, proportional gain, and integral gain, respectively. In order to simplify subsequent derivations, let us develop Process (1) into an infinite Taylor series [61]:

$$G_P(s) = \frac{A_0 - A_1 s + A_2 s^2 - A_3 s^3 + \cdots}{s}, \tag{8}$$

where $A_i$ represent the so-called "characteristic areas" of the process [60]. The areas can be expressed by process Parameters (1) in the following way [58]:

$$\begin{aligned} A_0 &= K_{PR} \\ A_1 &= K_{PR}\left(a_1 - b_1 + T_{delay}\right) \\ A_2 &= K_{PR}\left(b_2 - a_2 - T_{delay}b_1 + \frac{T_{delay}^2}{2!}\right) + A_1 a_1 \\ A_3 &= K_{PR}\left(a_3 - b_3 + T_{delay}b_2 - \frac{T_{delay}^2 b_1}{2!} + \frac{T_{delay}^3}{3!}\right) + A_2 a_1 - A_1 a_2. \\ &\quad \vdots \end{aligned} \tag{9}$$

By applying Expressions (7) and (1) to Expression (2), the following closed-loop transfer function parameters (5) are obtained:

$$\begin{aligned} e_0 &= A_0 K_i \\ e_1 &= -A_1 K_i + A_0 K_P \\ e_2 &= A_2 K_i - A_1 K_P + 1 \\ e_3 &= -A_3 K_i + A_2 K_P \\ e_4 &= A_4 K_i - A_3 K_P \\ &\quad \vdots \end{aligned} \tag{10}$$

$$\begin{aligned} f_0 &= A_0 K_i \\ f_1 &= -A_1 K_i + A_0 b K_P \\ f_2 &= A_2 K_i - A_1 b K_P \\ f_3 &= -A_3 K_i + A_2 b K_P, \\ &\quad \vdots \end{aligned} \tag{11}$$

In order to calculate two PI controller parameters, the first two equations in Conditions (6) should be satisfied. The first condition in (6) gives the following result:

$$K_i = 0.5A_0\left(1 - b^2\right)K_P{}^2, \tag{12}$$

and the second condition in (6) results in

$$K_i = \frac{\left(A_0 A_2 - 0.5A_1{}^2\right)\left(1 - b^2\right)K_P{}^2 \; + \; A_1 K_P - 0.5}{A_2}. \tag{13}$$

The controller gain ($K_P$) can be calculated by equating Expressions (12) and (13):

$$K_P = \frac{-A_1 + \sqrt{A_1{}^2 + \xi}}{\xi}, \tag{14}$$

where

$$\xi = \left(1 - b^2\right)\left(A_0 A_2 - A_1{}^2\right). \tag{15}$$

**Remark 1.** *If either reference weighting factor b is close or equal to 1, or the value of ξ in Equation (15) approaches zero, proportional gain $K_P$ can be calculated by developing the expression under the square root in Equation (14) into a Taylor series. In this case, the proportional gain becomes*

$$K_P = \frac{0.5}{A_1}. \tag{16}$$

*However, if b = 1 (1-DOF PI controller), integral Gain (12) becomes $K_i = 0$, and we obtain a proportional (P) controller. Therefore, disturbance-rejection performance is degraded when increasing the value of factor b. The influence of factor b on tracking and disturbance-rejection performance is shown through an illustrative example in Section 2.1.*

**Remark 2.** *While Expressions (12) and (14) result in stable and fast closed-loop responses for a large majority of IP models, closed-loop stability is still not guaranteed for an arbitrary process model. Closed-loop stability and robustness are discussed in detail in Section 3.*

Therefore, PI controller parameters can be expressed in terms of characteristic areas or process parameters by using Expression (9). However, characteristic areas can also be calculated from the process time response while changing the process steady state. To this end, the multiple integrations of the process input ($u(t)$) and output ($y(t)$) signals should be calculated [58,61]:

$$\begin{aligned}
u_0 &= \frac{u(t) - u(0)}{\Delta U} \\
I_{U1}(t) &= \int_0^t u_0(\tau)d\tau \quad I_{Y1}(t) = \frac{y(t) - y(0) - \dot{y}(0)\cdot t}{\Delta U}, \\
I_{U2}(t) &= \int_0^t I_{U1}(\tau)d\tau \quad I_{Y2}(t) = \int_0^t I_{Y1}(\tau)d\tau \\
&\vdots
\end{aligned} \tag{17}$$

where $u_0$ is normalised process input signal, and

$$\Delta U = u(\infty) - u(0). \tag{18}$$

The areas are expressed as follows:

$$
\begin{aligned}
A_0 &= y_0(\infty); \; y_1 = A_0 I_{U1}(t) - I_{Y1}(t) \\
A_1 &= y_1(\infty); \; y_2 = A_1 I_{U1}(t) - A_0 I_{U2}(t) + I_{Y2}(t) \\
A_2 &= y_2(\infty); \; y_3 = A_2 I_{U1}(t) - A_1 I_{U2}(t) + A_0 I_{U3}(t) - I_{Y3}(t), \\
&\;\vdots
\end{aligned}
\tag{19}
$$

where $\dddot{y}(0) = \ddddot{y}(0) = \cdots = 0$. In practice, integration time can be limited. The integration can be terminated when the process input and output signals in Equation (17) settle. Therefore, Equation (17) can be easily and recursively calculated in practice, and process Model (1) is not required [58,62,63].

The controller tuning steps are as follows:

- Determine characteristic areas from Expression (9) if the process model is known. Otherwise, modify the process steady state. During the process transient response, sample the controller and process output signals. Areas $A_0$ to $A_2$ are calculated from Expressions (17)–(19). The initial values of the process input and output signals can be estimated by averaging them before the change of the controller output signal.
- Choose appropriate reference weighting factor $0 \le b < 1$.
- Calculate controller parameters according to Equations (12) and (14).

All MATLAB and Simulink files required for controller parameters calculation are electronically accessible [63].

**Remark 3.** *Proportional gain $K_P$ for a second-order IP ($b_n = 0$, $a_3$-$a_n = 0$) can be calculated by the following expression:*

$$
K_P = \frac{-K_{PR}\left(a_1 + T_{delay}\right)\sqrt{K_{PR}{}^2\left(a_1 + T_{delay}\right)^2 + \xi}}{\xi},
\tag{20}
$$

*where*

$$
\xi = \frac{K_{PR}{}^2}{2}\left(b^2 - 1\right)\left(T_{delay}{}^2 + 2a_1 T_{delay} + 2a_2\right).
\tag{21}
$$

*Integral gain $K_i$ can be calculated from Expression (12) when taking into account that $A_0$ equals $K_{PR}$:*

$$
K_i = 0.5 K_{PR}\left(1 - b^2\right)K_P{}^2.
\tag{22}
$$

*When calculating PI controller parameters for a first-order process, $a_2$ in Expression (21) is replaced by 0.*

*2.1. Illustrative Example*

For the illustrative example, a second-order IP was selected:

$$
G_P(s) = \frac{1}{s(1+s)^2}.
\tag{23}
$$

Characteristic areas were calculated from Expression (9):

$$
A_0 = 1, \; A_1 = 2, \; A_2 = 3.
\tag{24}
$$

The PI controller parameters, presented in Table 1, were calculated from Expressions (12) and (14), or by Expressions (20) and (22). Parameters were calculated for different values of reference weighting factor $b$. The closed-loop responses for the input disturbance and for the reference change when applying a 2-DOF PI controller are shown in Figure 3.

**Table 1.** PI controller parameters.

| $b$ | $K_i$ | $K_P$ |
|-----|-------|-------|
| 0 | 0.0359 | 0.2679 |
| 0.2 | 0.0343 | 0.2671 |
| 0.5 | 0.0259 | 0.2630 |
| 0.7 | 0.0170 | 0.2585 |
| 0.9 | 0.0061 | 0.2530 |

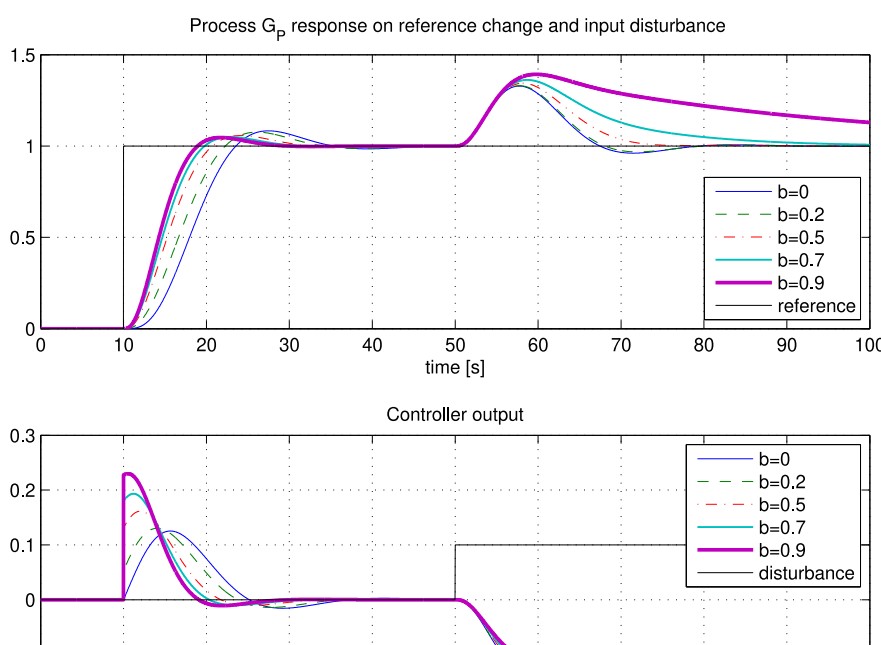

**Figure 3.** Process $G_P$ closed-loop responses. Comparison between different values of $b$.

As seen from Figure 3, tracking and disturbance-rejection performance are correlated with factor $b$. The reference following performance increases by increasing factor $b$. Therefore, if tracking performance is the most important, select a value of $b \geq 0.7$. On the other hand, the best disturbance-rejection performance is achieved with $b \leq 0.2$. A good compromise between reference following and disturbance-rejection performance is $b = 0.5$.

**Remark 4.** *The best overall response (optimal reference-tracking and disturbance-rejection) could be obtained by using a higher-order reference pre-filter instead of reference weighting factor b, similar to the proposed solution for non-integrating processes [64]. In this case, controller parameters should be the same as the ones calculated with b = 0, while pre-filter parameters should be adjusted so as to achieve the most optimal tracking response. However, such a solution would require calculating a relatively precise process model, and controller realisation would become much more complex, which is in contrast to the simplicity of the presented tuning method.*

## 3. Stability and Robustness

Stability can be assessed from closed-loop characteristic polynomial

$$1 + G_P G_{CY} = 0, \tag{25}$$

where process model $G_P$ is given by Expression (8), and controller $G_{CY}$ by Expression (7). It follows that

$$A_0K_I + s(A_0K_P - A_1K_I) + s^2(A_2K_I - A_1K_P + 1) + s^3(A_2K_P - A_3K_I) + s^4(A_4K_I - A_3K_P) + \cdots = 0. \quad (26)$$

Considering Condition (25), the necessary (but not sufficient) condition that all poles of the characteristic polynomial are in the left half plane is that coefficients of Equation (26) are positive. Consequently, the necessary conditions for closed-loop stability are

$$\begin{aligned}
A_0K_I &\geq 0 \\
A_0K_P - A_1K_I &\geq 0 \\
A_2K_I - A_1K_P &\geq -1 \\
(-1)^i(A_iK_I - A_{i-1}K_P) &\geq 0; \ i = 3, 4, \ldots
\end{aligned} \quad (27)$$

Sufficient stability conditions might be established by testing the Routh determinants. However, introducing pure time-delays into the process transfer function makes such analysis difficult or even impossible. Therefore, closed-loop stability and robustness were studied in detail on the following process model:

$$G(s) = \frac{K_{PR}e^{-T_{delay}s}}{s\left(1 + a_1 s + \alpha a_1^2 s^2\right)}. \quad (28)$$

The process ($G(s)$) represents the first- or second-order integrating system with or without time-delay. The stability and robustness of the closed-loop system when applying the proposed tuning method depend on the ratio between parameter $a_1$ and $T_{delay}$, parameter $\alpha$, and on chosen parameter $b$. Processes with real poles have $\alpha \leq 0.25$.

The closed-loop system is stable for any chosen time-delay and controller parameter $b$ (between 0 and 1) if ratio $\alpha < 0.869$. Therefore, the closed-loop response is stable for all processes with real poles, or with moderate complex conjugate poles (up to $\alpha < 0.869$). Stability improves by increasing time-delay and controller parameter $b$ (see Figure 4).

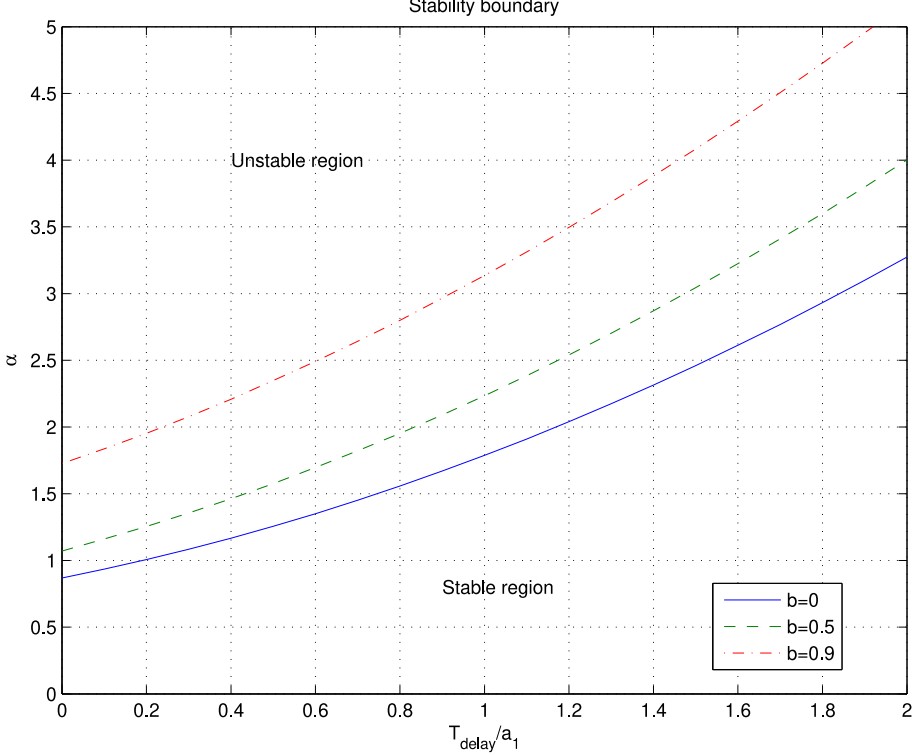

**Figure 4.** Stability margin for process $G(s)$ at different values of controller parameter $b$.

The robustness of the tuning method was evaluated by measuring the maximal sensitivity of the system ($M_s$) [1]. Parameter $M_s$ represents the inverse of the shortest distance of open-loop transfer function $G_P G_{CY}$ Nyquist curve to critical point –1. Larger $M_s$ values denote a less stable system. Usual design values for $M_s$ are in the range of 1.4–2 [1].

$M_s$ values, obtained for controller parameters calculated at $b = 0$, $b = 0.5$, and $b = 0.9$, are shown in Figures 5–7, respectively. For processes with real poles ($\alpha \leq 0.25$), expected $M_s$ values were approximately between 1.7 and 2.3 for $b = 0$, between 1.5 and 1.9 for $b = 0.5$, and between 1.3 and 1.7 for $b = 0.9$ (upper values of $M_s$ were obtained for highly delayed processes). Expected closed-loop robustness is therefore relatively high for a variety of process models.

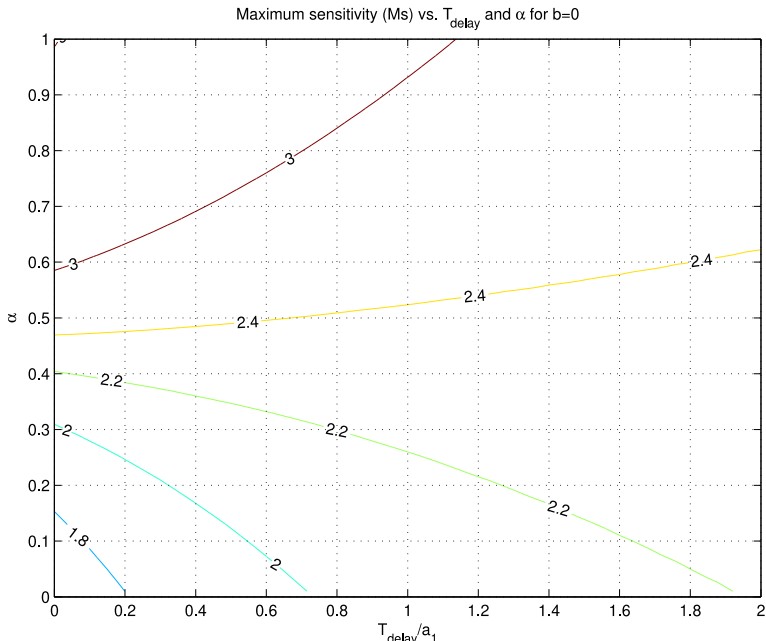

**Figure 5.** Closed-loop maximum sensitivity ($M_S$) for $G(s)$ and controller parameter $b = 0$ at different values of process parameters $T_{delay}$ and $\alpha$.

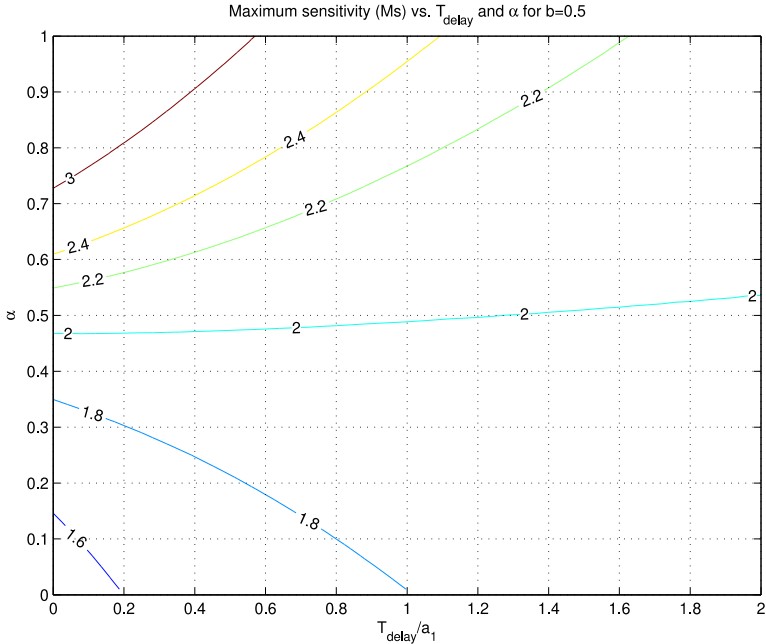

**Figure 6.** Closed-loop maximum sensitivity ($M_S$) for $G(s)$ and controller parameter $b = 0.5$ at different values of process parameters $T_{delay}$ and $\alpha$.

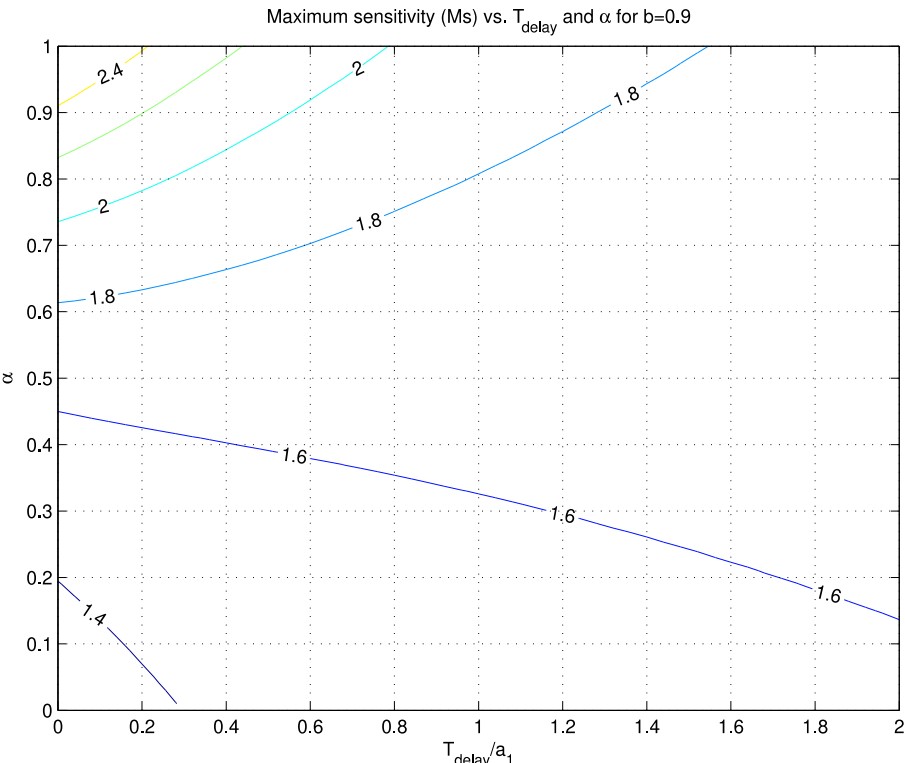

**Figure 7.** Closed-loop maximum sensitivity ($M_S$) for $G(s)$ and controller parameter $b = 0.9$ at different values of process parameters $T_{delay}$ and $\alpha$.

## 4. Examples and Comparisons

The proposed method was tested on four process models and compared to some other methods.

### 4.1. Experiments on Process Models

The following second-order, delayed, fourth-order, and non-minimum phase IPs were chosen:

$$
\begin{aligned}
G_{P1}(s) &= \frac{1}{s(1+s)(1+0.2s)} \\
G_{P2}(s) &= \frac{e^{-s}}{s} \\
G_{P3}(s) &= \frac{1}{s(1+s)^4} \\
G_{P4}(s) &= \frac{1-2s}{s(1+s)^2}.
\end{aligned}
\tag{29}
$$

The calculated areas, according to Expression (9), are given below:

$$
\begin{aligned}
G_{P1} &: A_0 = 1,\ A_1 = 1.2,\ A_2 = 1.24 \\
G_{P2} &: A_0 = 1,\ A_1 = 1,\ A_2 = 0.5 \\
G_{P3} &: A_0 = 1,\ A_1 = 4,\ A_2 = 10 \\
G_{P4} &: A_0 = 1,\ A_1 = 4,\ A_2 = 7.
\end{aligned}
\tag{30}
$$

The PI controller parameters for all four process models, for $b = 0$, $b = 0.5$, and $b = 0.9$, were calculated from Equations (12) and (14), and are given in Table 2.

**Table 2.** PI controller parameters for different process models and parameter $b$.

| | $b = 0$ | | $b = 0.5$ | | $b = 0.9$ | |
|---|---|---|---|---|---|---|
| | $K_i$ | $K_p$ | $K_i$ | $K_p$ | $K_i$ | $K_p$ |
| $G_{P1}$ | 0.093 | 0.432 | 0.069 | 0.428 | 0.017 | 0.419 |
| $G_{P2}$ | 0.172 | 0.586 | 0.117 | 0.559 | 0.025 | 0.512 |
| $G_{P3}$ | 0.0097 | 0.140 | 0.0069 | 0.135 | 0.0015 | 0.127 |
| $G_{P4}$ | 0.0113 | 0.151 | 0.0076 | 0.142 | 0.0016 | 0.128 |

The closed-loop responses when applying a 2-DOF PI controller with three values of $b$ for input disturbance ($d = 0.1$) and for reference change are shown in Figures 8–11. Experiment results on all four process models showed that the proposed tuning method gave relatively fast and highly damped responses with slight overshoots, in accordance with the magnitude optimum criterion. An exception was for $b = 0.9$, where slow disturbance-rejection responses could be observed (see Section 2.1).

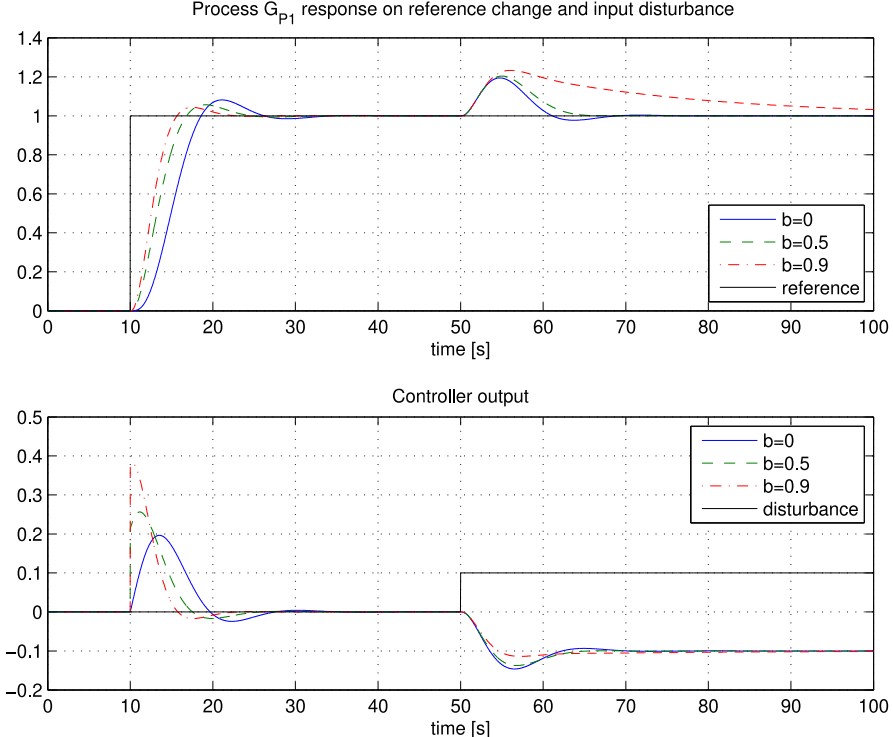

**Figure 8.** $G_{P1}$ process closed-loop experiment.

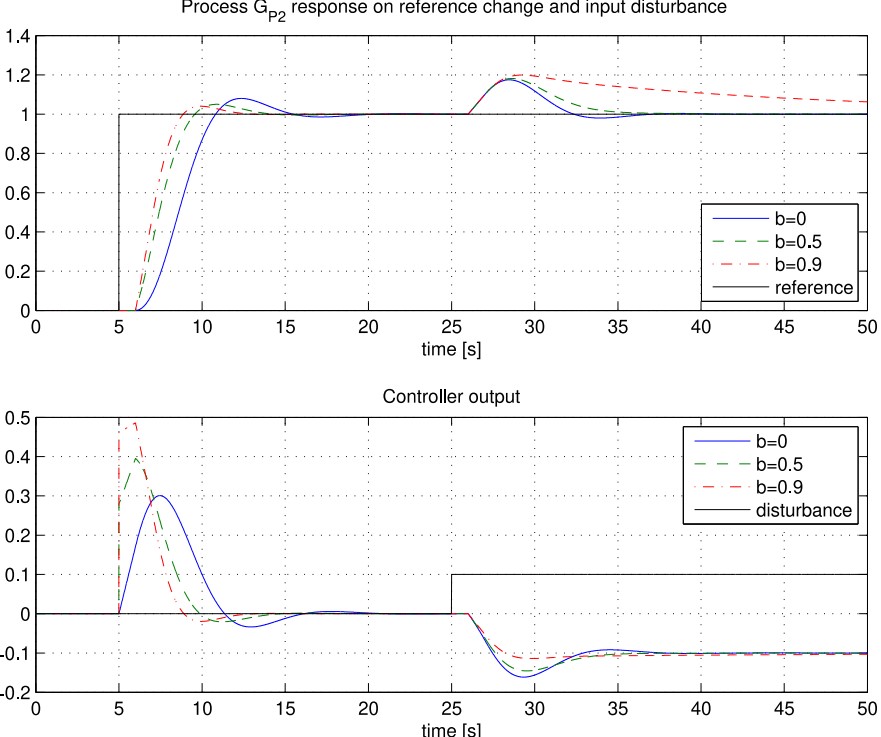

**Figure 9.** $G_{P2}$ process closed-loop experiment.

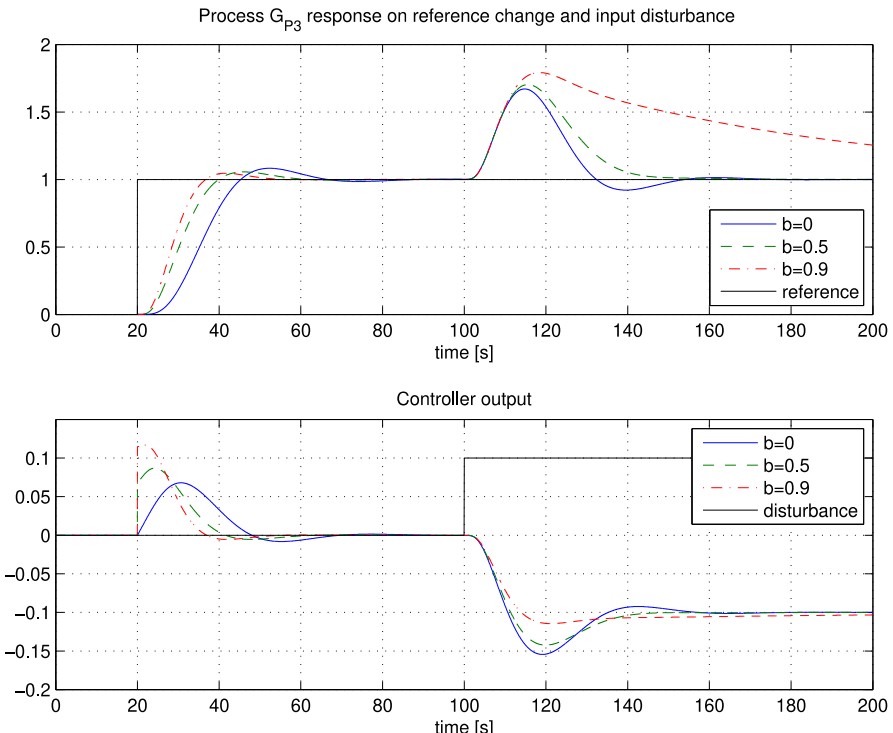

**Figure 10.** $G_{P3}$ process closed-loop experiment.

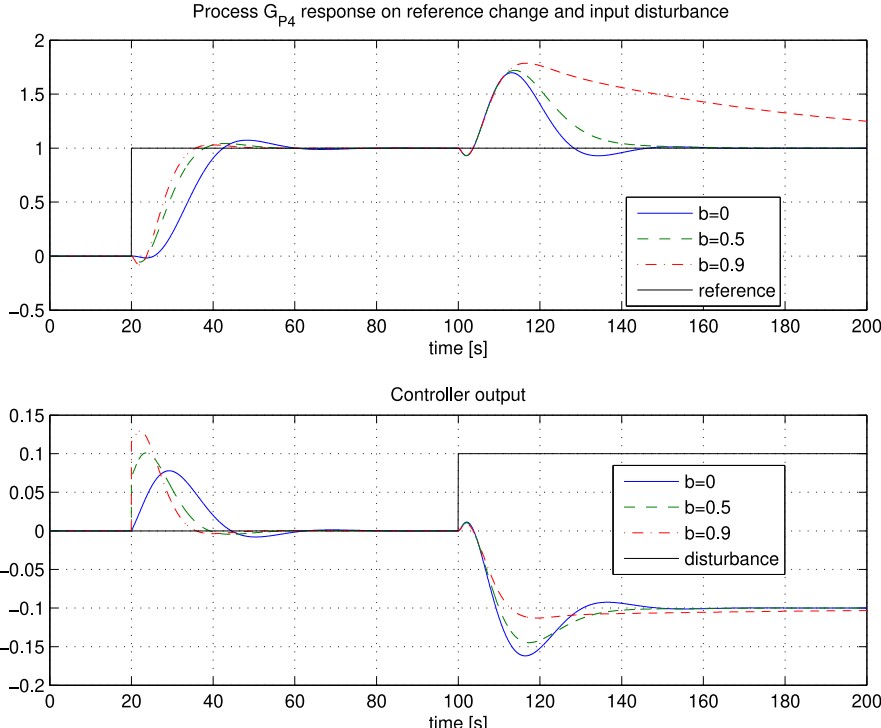

**Figure 11.** $G_{P4}$ process closed-loop experiment.

The closed-loop amplitude response (between reference and process output) is shown in Figure 12. It is clear that all four processes, at different selections of parameter *b*, gave a flat amplitude response at lower frequencies, all according to the MO criterion (see Figure 2).

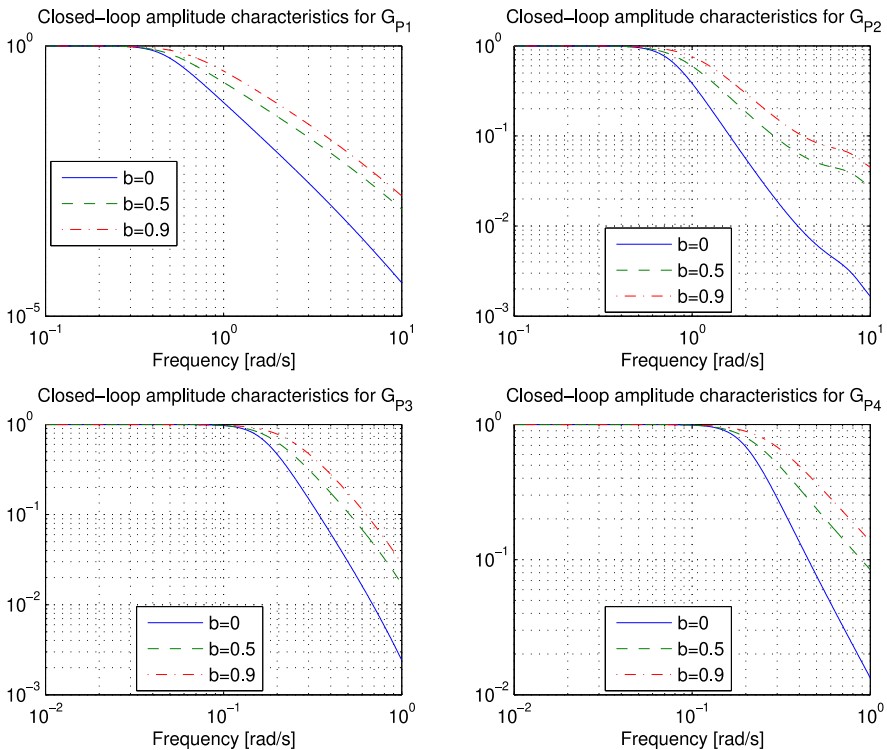

**Figure 12.** Closed-loop amplitude responses for all four tested processes.

## 4.2. Comparisons with Other Methods

To the best of our knowledge, no other tuning method for IP offers both tuning based on a general process transfer function with arbitrary order with time-delay, and on the measurement of a simple process open- or closed-loop time response in the time-domain. In practical applications, this represents a substantial advantage over other tuning methods. However, the proposed tuning method was compared to some other existing tuning methods for PI controller. Selected methods are suitable for IP and employed a relatively simple experiment on the process to calculate the controller parameters. We compared four different IP models that are frequently encountered in the literature.

The reference-tracking (*r* = 1) and disturbance-rejection responses (*d* = 0.1) were measured on all processes. Tracking and disturbance-rejection performances were evaluated by the following criterion:

$$ITSE = \int_0^\infty t \cdot e^2(t)dt, \tag{31}$$

where *e(t)* is control error (*r-y*). The integral-time-square-error (ITSE) criterion favours fast and non-oscillatory responses. Tracking response was additionally evaluated by measuring 5% settling time. In the proposed tuning method, controller parameter *b* = 0.5 was used in all four cases. This represents a trade-off between tracking and disturbance-rejection performance, and the reference-tracking or disturbance-rejection performance of the proposed method can be additionally improved/adjusted by changing parameter *b*.

### Case 1

The first comparison was performed on the first-order process with a relatively small delay:

$$G_{P1}(s) = \frac{e^{-0.05s}}{s(1+s)}. \tag{32}$$

On the basis of the given process model, areas were determined from Equation (9):

$$A_0 = 1, \ A_1 = 1.05, \ A_2 = 1.0513. \tag{33}$$

Then, the controller was designed according to Equations (12) and (14):

$$K_P = 0.48, \ K_I = 0.087, \ b = 0.5. \tag{34}$$

The obtained controller was compared to three other methods [1,37,65]. The first one was Åström's method [1]. This method is based on choosing maximal sensitivity value Ms = 1.4 (*Ms*14) or Ms = 2.0 (*Ms*20). The second method was that of Taguchi and Araki (TA) [65]. This method calculates the controller parameters in order to optimize settling time while keeping overshoot below 20%. The third method was that of Ali and Majhi (ALI) [37]. This method optimises the integral-square-error (ISE) criterion and the shape of the Nyquist curve.

The calculated PI controller parameters for the presented methods are given in Table 3.

**Table 3.** PI controller parameters for Case 1.

| Parameter | Ms14 | Ms20 | TA | ALI |
|:---:|:---:|:---:|:---:|:---:|
| $K_P$ | 0.386 | 0.733 | 5.67 | 12.69 |
| $K_I$ | 0.06 | 0.203 | 1.26 | 2.14 |
| $b$ | 0.37 | 0.71 | 0.34 | 1 |

Figure 13 compares closed-loop responses for tracking and for disturbance-rejection between the mentioned tuning rules applied to 2-DOF PI controllers. The values of the ITSE criterion for all tested methods are given in Table 4 (for reference tracking) and Table 5 (for disturbance-rejection).

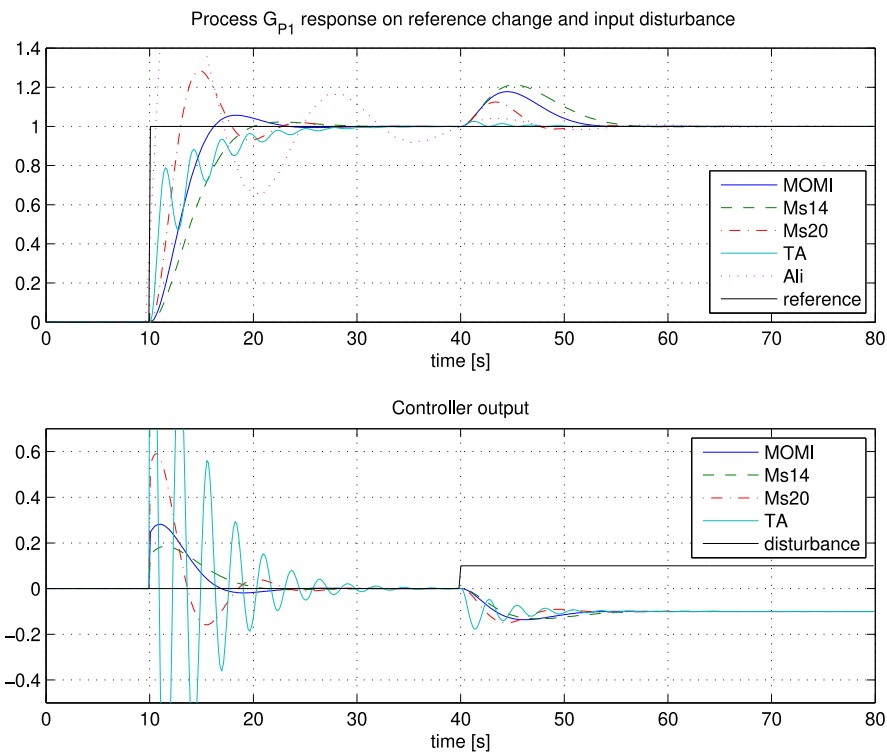

**Figure 13.** Closed-loop responses of $G_{P1}$ process. PI controllers were tuned by the magnitude optimum multiple integration (MOMI), Ms14, Ms20, TA, and ALI methods. For clarity, the ALI response is not shown in the bottom picture.

**Table 4.** ITSE criterion values for reference tracking.

| Process | MOMI | MS14 | MS20 | TA | ALI |
|---------|------|------|------|------|------|
| $G_{P1}$ | 3.06 | 5.71 | 2.15 | 2.31 | 13.63 |
| $G_{P2}$ | 2.27 | 8.12 | 2.61 | 2.33 | 4.66 |
| $G_{P3}$ | 21.44 | 59.59 | 27.71 | 19.82 | - |
| $G_{P4}$ | 9.39 | 25.95 | 14.45 | - | - |

**Table 5.** ITSE criterion values for disturbance-rejection.

| Process | MOMI | MS14 | MS20 | TA | ALI |
|---------|------|------|------|------|------|
| $G_{P1}$ | 0.76 | 1.57 | 0.18 | 0.0025 | 0.0014 |
| $G_{P2}$ | 0.45 | 8.05 | 0.39 | 0.20 | 0.71 |
| $G_{P3}$ | 39.4 | 624.0 | 59.3 | 11.56 | - |
| $G_{P4}$ | 7.51 | 165.5 | 14.83 | - | - |

Tracking settling times within ±5% of the final value were comparable for the first four methods (see Table 6), where MOMI ranked second, just after Ms14. The Ms20 method showed a somewhat large overshoot, while the TA and ALI methods exhibited an oscillatory response. According to the ITSE criterion, reference-tracking performance (Table 4) was the best with the TA method. The lowest ITSE criterion for disturbance-rejection was obtained with the ALI method (measurement of disturbance-rejection was taken after the process output completely settled). However, due to a relatively high degree of oscillations, neither the TA nor the ALI method could be considered as the most appropriate for the $G_{P1}$ process.

**Table 6.** The 5% settling times for reference tracking.

| Process | MOMI | MS14 | MS20 | TA | ALI |
|---------|------|------|------|------|------|
| $G_{P1}$ | 9.4 | 8.5 | 11.0 | 11.6 | 30.5 |
| $G_{P2}$ | 6.1 | 21.7 | 4.7 | 8.8 | 11.3 |
| $G_{P3}$ | 21.3 | 35.3 | 30.3 | 27.0 | - |
| $G_{P4}$ | 13.5 | 23.2 | 21.3 | - | - |

The proposed method response was between methods Ms20 and Ms14, faster than Ms20 but slower than Ms14.

**Case 2**

The second comparison was performed on the process with pure time-delay

$$G_{P2}(s) = \frac{e^{-s}}{s}. \tag{35}$$

On the basis of the given process model, areas were determined from Expression (9):

$$A_0 = 1, \; A_1 = 1, A_2 = 0.5. \tag{36}$$

Then, the controller was designed according to Equations (12) and (14):

$$K_P = 0.56, \; K_I = 0.12, \; b = 0.5. \tag{37}$$

Once more, the proposed MOMI method was compared to the presented tuning methods. The calculated PI controller parameters for the presented methods are given in Table 7.

**Table 7.** PI controller parameters for Case 2.

| Parameter | Ms14 | Ms20 | TA | ALI |
|-----------|------|------|------|------|
| $K_P$ | 0.332 | 0.577 | 0.766 | 0.48 |
| $K_I$ | 0.021 | 0.129 | 0.187 | 0.092 |
| $b$ | 0.6 | 0.39 | 0.32 | 1 |

Figure 14 compares the closed-loop responses for tracking and disturbance-rejection between the mentioned tuning rules applied to 2-DOF PI controllers.

The shortest 5% settling time was obtained with the Ms20 method. The ITSE criterion for tracking performance revealed that the MOMI, TA, and Ms20 methods gave almost identical results, while Ms14 and ALI gave noticeably higher values. The TA method exhibited some undershoot, and the ALI method some overshoot. For disturbance-rejection performance, ITSE favoured the TA method, while Ms20 and MOMI were close to each other (MOMI ranked third).

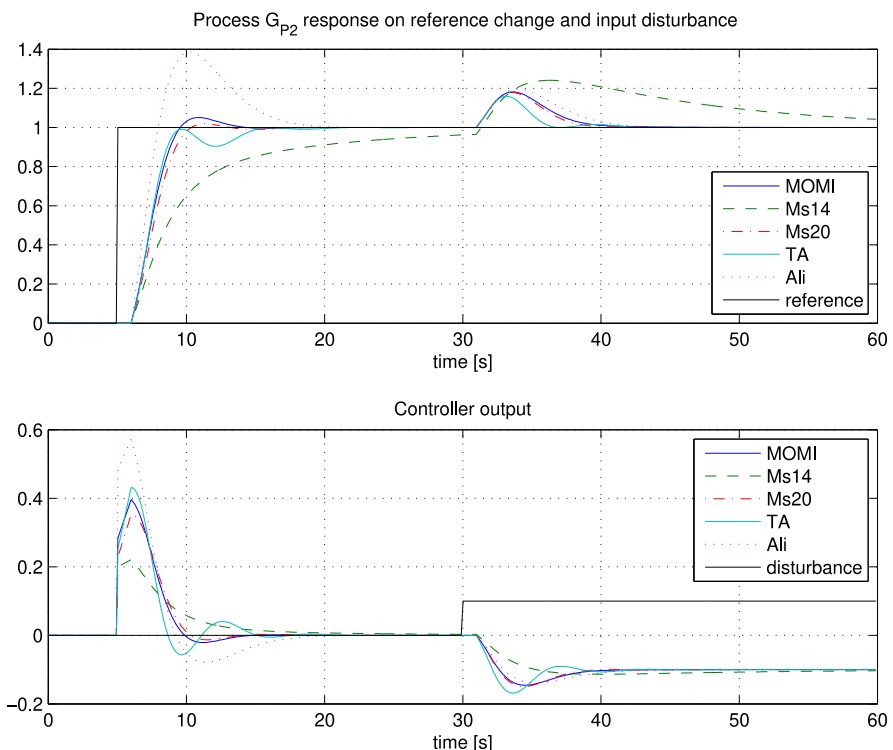

**Figure 14.** Closed-loop responses of $G_{P2}$ process. PI controllers were tuned by MOMI, Ms14, Ms20, TA, and ALI methods.

**Case 3**

The third comparison was performed on second-order process with time-delay

$$G_{P3}(s) = \frac{e^{-s}}{s(1+s)^2}.\tag{38}$$

On the basis of the given process model, areas were determined from Equation (9):

$$A_0 = 1,\ A_1 = 3,\ A_2 = 5.5.\tag{39}$$

Then, the controller was designed according to Equations (12) and (14):

$$K_P = 0.181, \ K_I = 0.012, \ b = 0.5. \tag{40}$$

Once more, the proposed MOMI method was compared with some of the presented tuning methods. The calculated PI controller parameters for the presented methods are given in Table 8.

**Table 8.** PI controller parameters for Case 3.

| Parameter | Ms14 | Ms20 | TA |
|---|---|---|---|
| $K_P$ | 0.095 | 0.154 | 0.291 |
| $K_I$ | 0.0026 | 0.0104 | 0.023 |
| $b$ | 0.6 | 0.48 | 0.32 |

Figure 15 compares closed-loop responses for tracking and for disturbance-rejection between mentioned tuning rules applied to 2-DOF PI controllers.

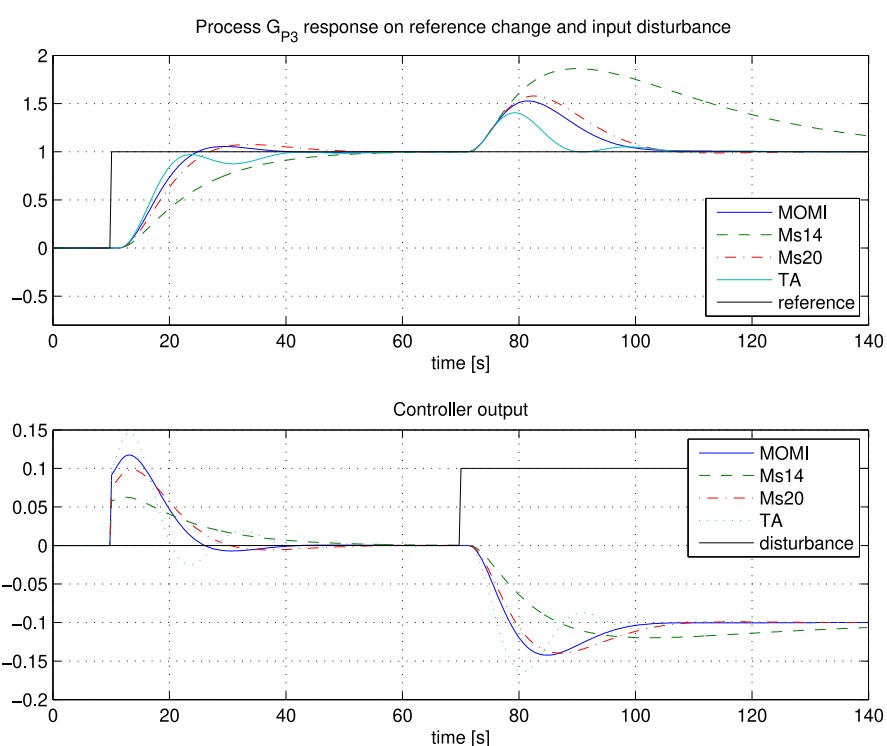

**Figure 15.** Closed-loop responses of $G_{P3}$ process. PI controllers were tuned by MOMI, Ms14, Ms20, and TA methods.

The fastest 5% settling time for reference tracking was obtained by the MOMI method. The ITSE criterion for reference tracking (see Table 4) gave the best value for the TA method, but the MOMI method remained close. Again, the TA method had some undershoot. Similar results were obtained with the ITSE criterion for disturbance-rejection (Table 5).

**Case 4**

The last comparison was performed on high-order process

$$G_{P4}(s) = \frac{1}{s(1+0.25s)^8}. \tag{41}$$

On the basis of the given process model, areas were determined from Equation (9):

$$A_0 = 1, \ A_1 = 2, \ A_2 = 2.25. \tag{42}$$

Then, the controller was designed according to Equations (12) and (14):

$$K_P = 0.27, \ K_I = 0.028, \ b = 0.5. \tag{43}$$

Once more, the proposed MOMI method was compared with some of the presented tuning methods. The calculated PI controller parameters for the presented methods are given in Table 9.

**Table 9.** PI controller parameters for Case 4.

| Parameter | Ms14 | Ms20 |
|:---:|:---:|:---:|
| $K_P$ | 0.136 | 0.215 |
| $K_I$ | 0.005 | 0.021 |
| $b$ | 0.66 | 0.45 |

Figure 16 compares closed-loop responses for tracking and disturbance-rejection between the mentioned tuning rules applied to 2-DOF PI controllers.

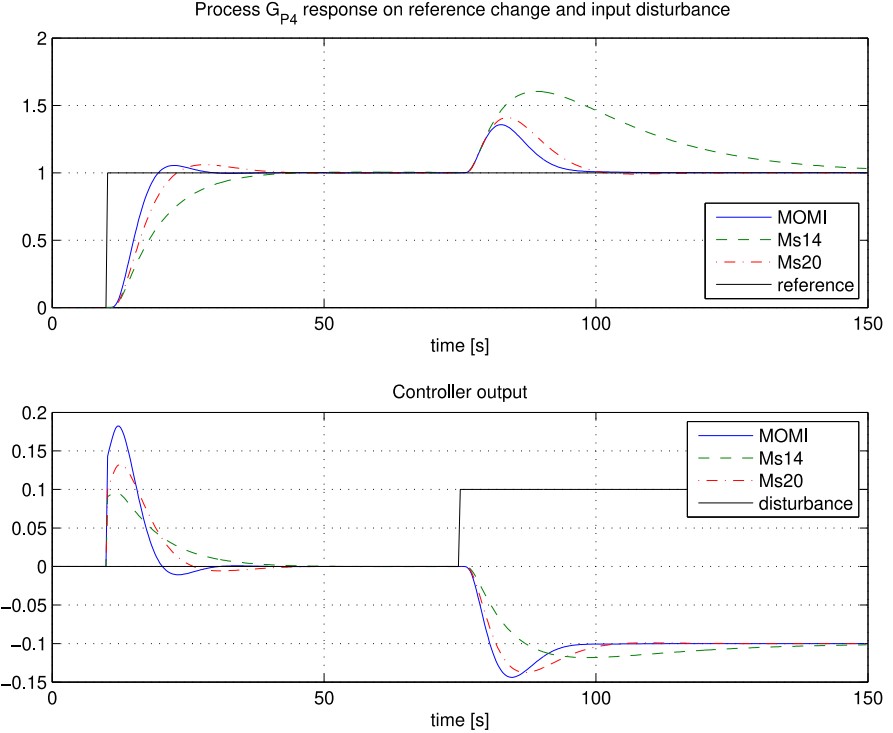

**Figure 16.** Closed-loop responses of $G_{P4}$ process. PI controllers were tuned by MOMI, Ms14, and Ms20 methods.

The fastest 5% settling time for reference change was obtained by the MOMI method. Similarly, the MOMI method gave the lowest ITSE values for reference change and disturbance-rejection.

Experiment results showed that the proposed MOMI method gave very good closed-loop responses for a broad range of process models when compared to some other methods. The disturbance-rejection and tracking responses were relatively fast and stable, with relatively small overshoots. As mentioned before, the obtained tracking or disturbance-rejection performance can be additionally improved by appropriately modifying controller parameter $b$.

## 5. Real-Time Experiments

### 5.1. Experiment on Control System for Charge-Amplifier Drift Compensation

The proposed tuning method was also tested on control system for charge-amplifier drift compensation that was part of a custom-made measuring system for the automated low-frequency and high-temperature polarization measurements of dielectric materials [66]. The measurement system consisted of a temperature-regulated furnace (heating of the measured samples), lock-in amplifier Stanford Research Systems SR830, and commercial charge-amplifier Kistler 5018A. This dielectric-material characterisation concept is presented in [67,68].

The control process (charge amplifier) had an integrating character. Process input and output were voltage signals limited from −10 to 10 V. The 2-DOF PI controller was realized with software package LabVIEW, and data acquisition was realized with an NI USB-6001 card. More information about this particular control system for drift compensation can be found in [66].

First, a step-change at the process input (change of input voltage signal from 0 to −2 V) was applied. The process response (voltage) is shown in Figure 17. From the measured open-loop responses, the following characteristic areas were calculated using Expressions (17)–(19):

$$A_0 = 0.1119, \ A_1 = 0.009, \ A_2 = 5.538 \times 10^{-4}. \tag{44}$$

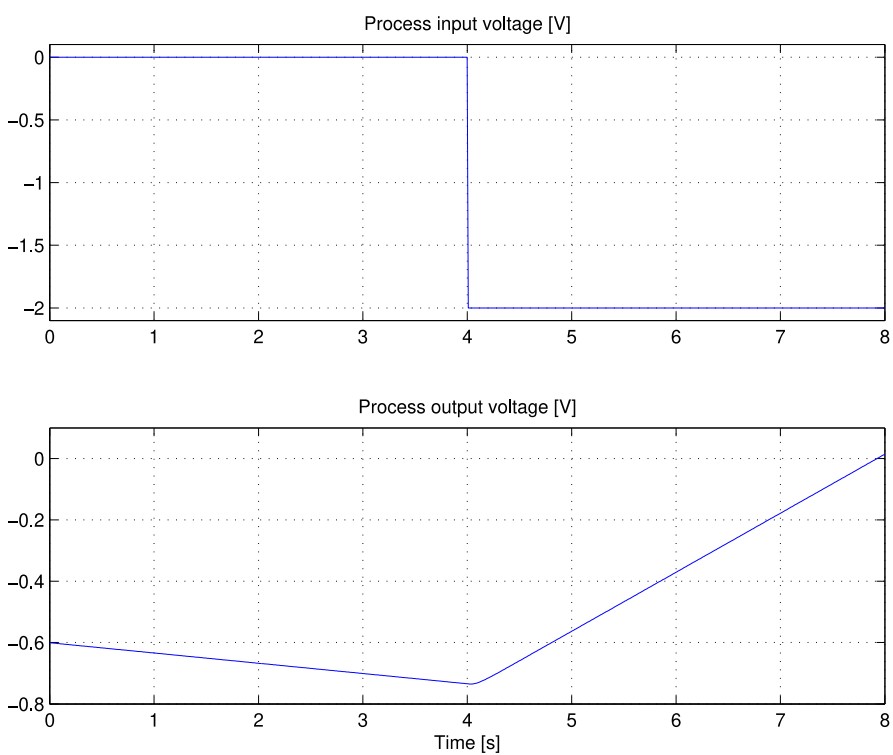

**Figure 17.** Open-loop response of control system for charge-amplifier drift compensation.

Using Expressions (12) and (14), the PI controller parameters for different values of reference weighting factor *b* (0, 0.5, and 0.7) were calculated. The calculated parameters are presented in Table 10.

**Table 10.** PI controller parameters for different values of factor $b$.

| $b$ | $K_i$ | $K_P$ |
|-----|-------|-------|
| 0 | −196.1 | −59.2 |
| 0.5 | −141.9 | −58.17 |
| 0.7 | −93.4 | −57.2 |

Figure 18 shows the process closed-loop response on a reference change, and artificially added step-disturbance ($d = 1$) at the process input ($d$). According to the responses, it is evident that the proposed method results in quite efficient control of the drift compensation of the charge amplifier. Reference following performance increases by increasing factor $b$, and the best disturbance-rejection performance is achieved with $b = 0$. A good compromise between disturbance-rejection and reference following performance is $b = 0.5$.

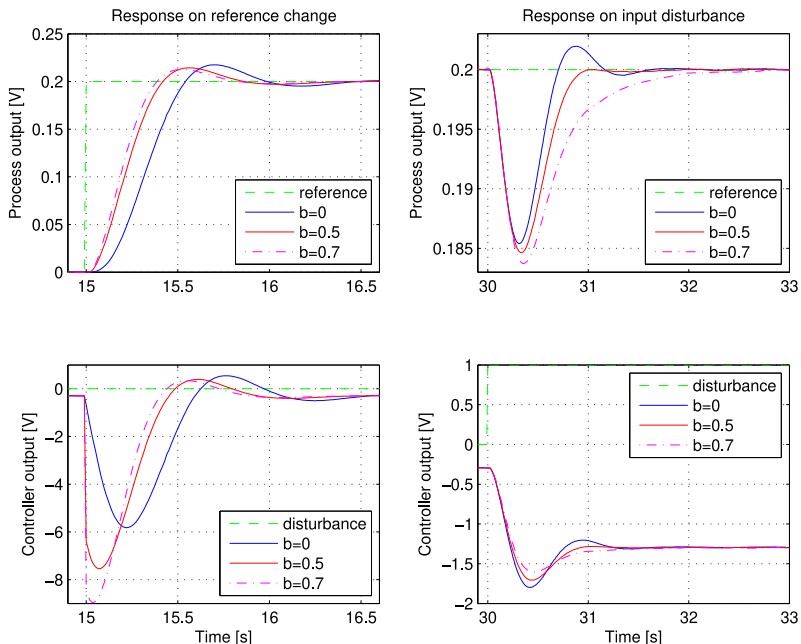

**Figure 18.** Closed-loop response of control system for charge-amplifier drift compensation.

## 5.2. Experiment on Hydraulic Laboratory Plant

The proposed tuning method was tested on a hydraulic laboratory plant, as shown in Figure 19. The plant consisted of three water tanks. Figure 20 presents a schematic diagram of the setup. The process input was voltage on pump $P_1$, and the process output was a filtered-water level in the first tank ($h_1$). Valve $V_{12}$ was closed (see Figure 20). Since water was coming out of tank $T_1$ to the main reservoir placed below three tanks (valve $V_1$ was open), the process did not contain an integrator. However, since the main time constant was very large, the process could be considered as an approximation of an integrating system. Water levels were measured with pressure sensors. For data acquisition, a custom-made USB-DAQ card was used, and the closed-loop control was implemented in MATLAB and Simulink software packages. More information about this particular laboratory plant can be found in [69].

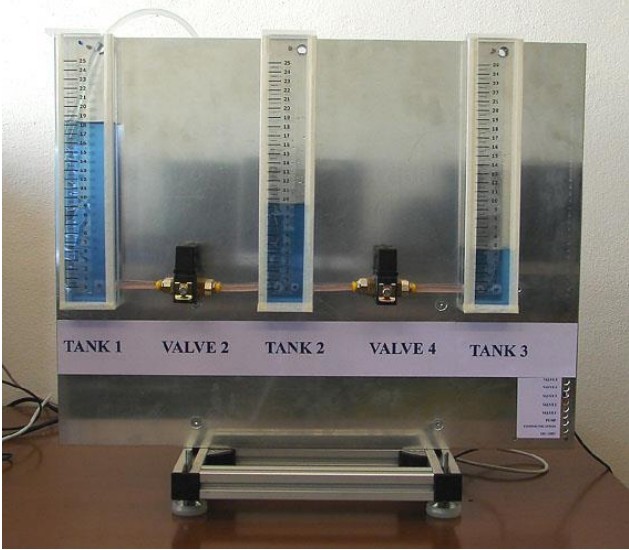

**Figure 19.** Hydraulic laboratory setup.

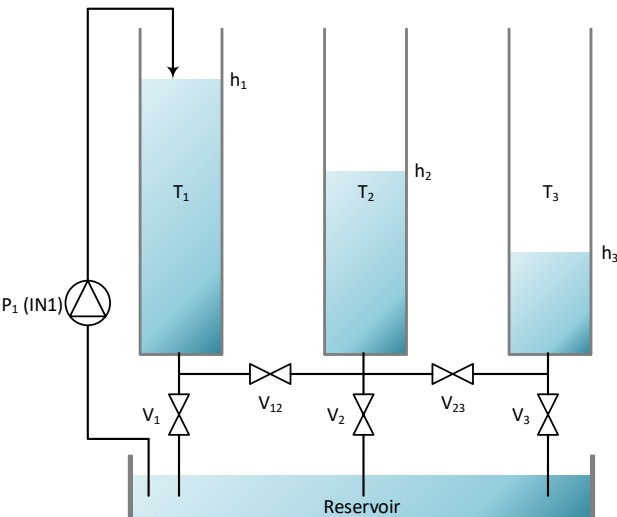

**Figure 20.** Schematic diagram of hydraulic laboratory setup.

The open- and closed-loop experiments were performed on the setup. First, voltage step-change was applied to pump $P_1$. Voltage (process input) and water level in reservoir $T_1$ (process output) are shown in Figure 21. The actual water level (in mm) could be obtained by multiplying the measured process output value by 0.5.

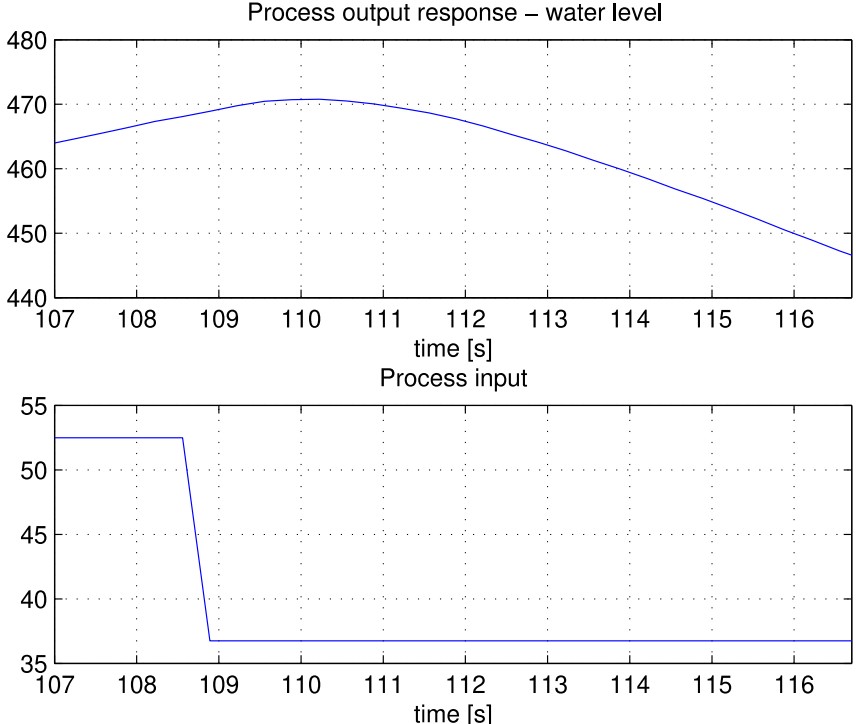

**Figure 21.** Process input (voltage on pump in %) and output (water level in mm multiplied by a factor of 2) signals during manual experiment on hydraulic process.

From the measured open-loop responses, the following characteristic areas were calculated by using Expressions (17)–(19):

$$A_0 = 0.476, \; A_1 = 0.994, \; A_2 = 1.57. \tag{45}$$

Using Expressions (12) and (14), the PI controller parameters were calculated:

$$K_I = 0.069, \; K_P = 0.54, \tag{46}$$

where, to improve disturbance-rejection performance, factor $b = 0$ was chosen ($b = 0$ corresponds to a 2-DOF controller).

The closed-loop responses on two reference changes and artificially added step-disturbances at process input ($d$) are shown in Figure 22. According to the response in Figure 22, it is evident that the proposed method results in a quite efficient control of the hydraulic plant. The nonlinear behaviour of the plant could be seen from different responses on positive and negative references, and input step-disturbance changes.

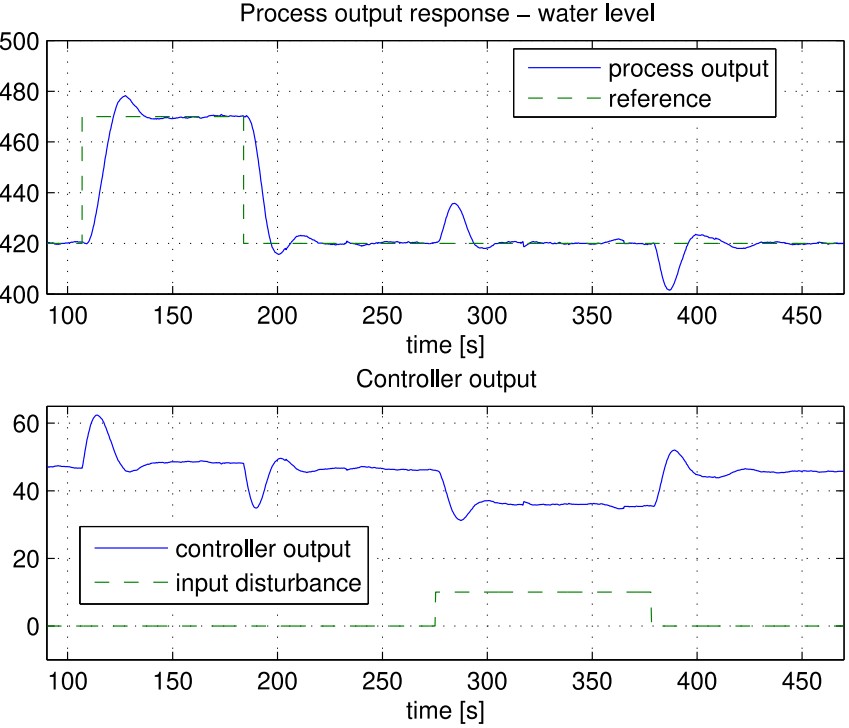

**Figure 22.** Process input (voltage on pump in %) and output (water level in mm multiplied by a factor of 2) signals during closed-loop experiment on hydraulic process.

## 5.3. Experiment on Industrial Autoclave

The new tuning method was also tested on an industrial autoclave, as shown in Figure 23. An autoclave is used in the production of plastics. Process input is the electric power on heaters (from 0 to 100%), and process output is inner temperature (in °C). The process was supervised by a SCADA system connected to the underlying PLC controllers with a sample time of 1 s.

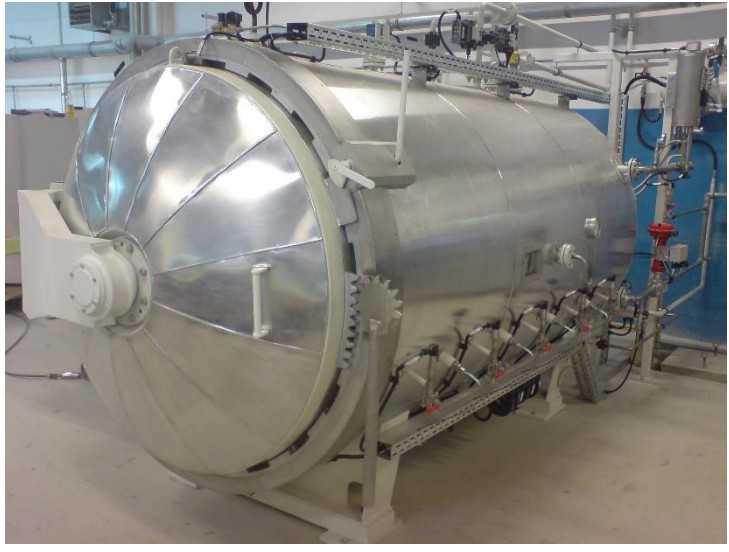

**Figure 23.** Industrial autoclave.

The process was self-regulatory with a main time constant of more than 1 day. However, the desired closed-loop time constant was less than 30 min. On the other hand, in the first few minutes,

the process open-loop step-response was similar to the IP with delay (see Figure 24). Therefore, for the calculation of the controller parameters, only the first part of the process open-loop response was used.

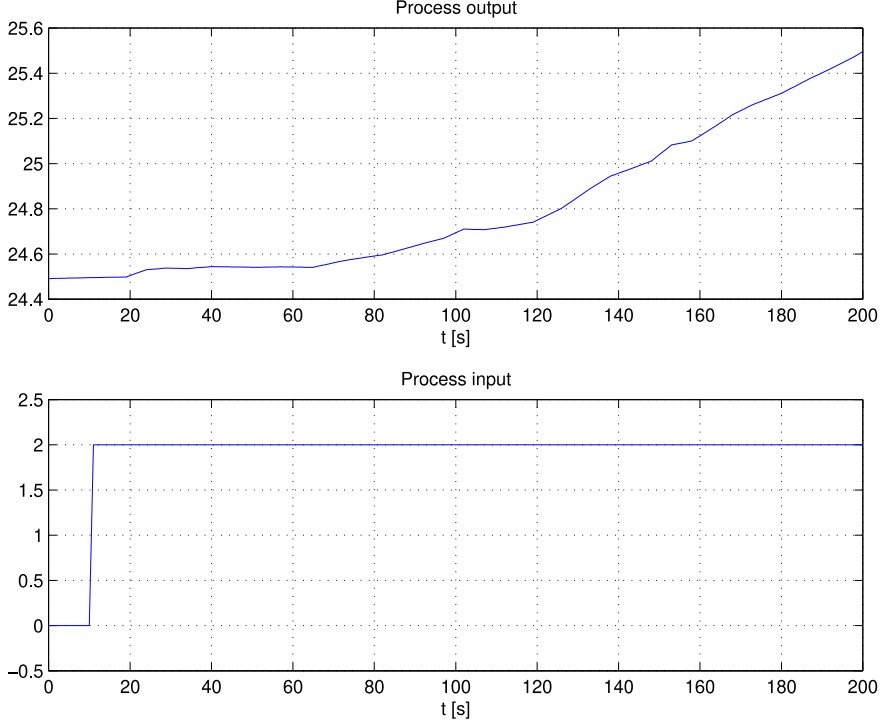

**Figure 24.** Process input (electric power in %) and output (temperature in °C) signals during manual experiment on autoclave within first 3 min.

First, a step-change at the process input (change of electric power from 0% to 2%) was applied. Figure 24 shows the process response (temperature). From the measured open-loop responses, using Expressions (17)–(19), the following characteristic areas were calculated:

$$A_0 = 0.0044, \; A_1 = 0.3651, \; A_2 = 17.86. \tag{47}$$

Using Expressions (12) and (14), the PI controller parameters were calculated:

$$K_I = 0.052, \; K_P = 1.55, \tag{48}$$

where factor $b = 0$ was chosen in order to improve tracking performance.

Figure 25 shows the closed-loop responses on a reference change. From the obtained closed-loop responses, it is evident that the proposed method results in quite efficient control of the autoclave temperature.

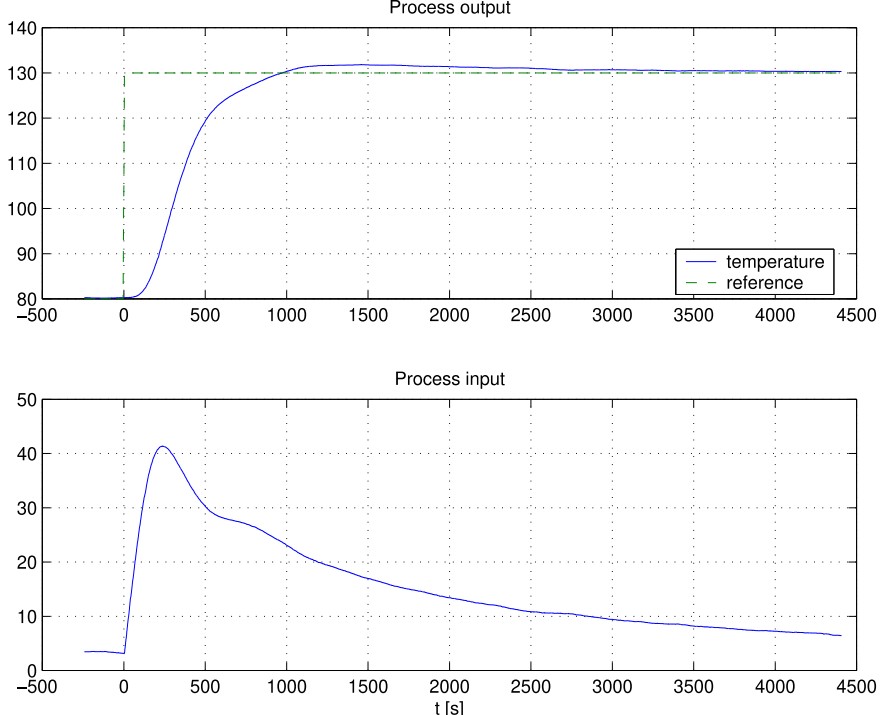

**Figure 25.** Process input and output signals during closed-loop autoclave experiment.

### 5.4. Experiment on Solid-Oxide Fuel Cell

The proposed tuning method was also tested on the temperature control of an air heat exchanger of a 10 kW solid-oxide fuel cell (SOFC) that comprised 80 cells. The process input was the flow of natural gas (NG) in standard litres per minute. The process output was the temperature of the heat exchanger in degrees Celsius. Control was performed with a Mitsubishi PLC controller [70] with a sample time of 1 s.

The open-loop measurements are shown in Figure 26. Available measurements were not long enough to find the process model. However, since the process, at the beginning of its response (between 0 and 900 s), resembles the IP with delay, the controller parameters were calculated with the proposed tuning method.

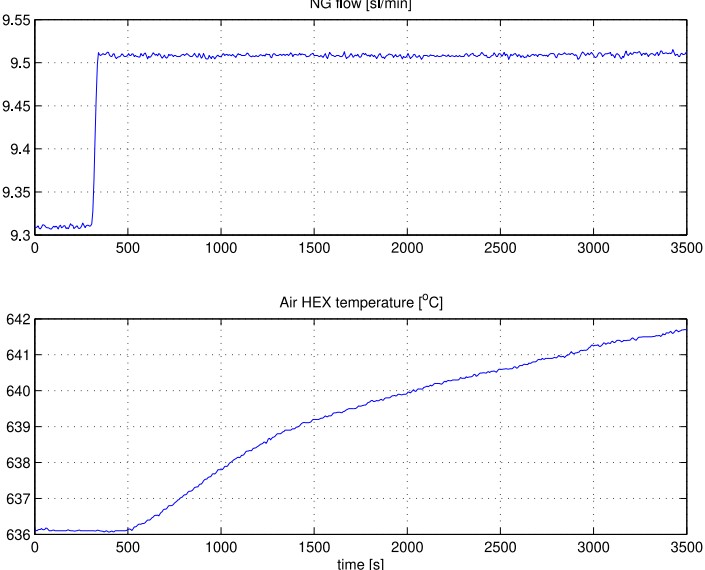

**Figure 26.** Open-loop response of Air HEX control loop.

The calculated areas were

$$A_0 = 0.0204, \ A_1 = 4.667, \ A_2 = 610.96. \tag{49}$$

The calculated controller parameters were

$$K_P = 0.122, \ K_i = 1.517 \cdot 10^{-4}, \ T_F = 804 \ s, \tag{50}$$

where $T_F$ stands for the first-order filter placed in front of the reference signal.

The closed-loop experiment was performed by increasing the Air HEX reference temperature from 652 to 642 °C; Figure 27 shows the closed-loop response is stable and smooth.

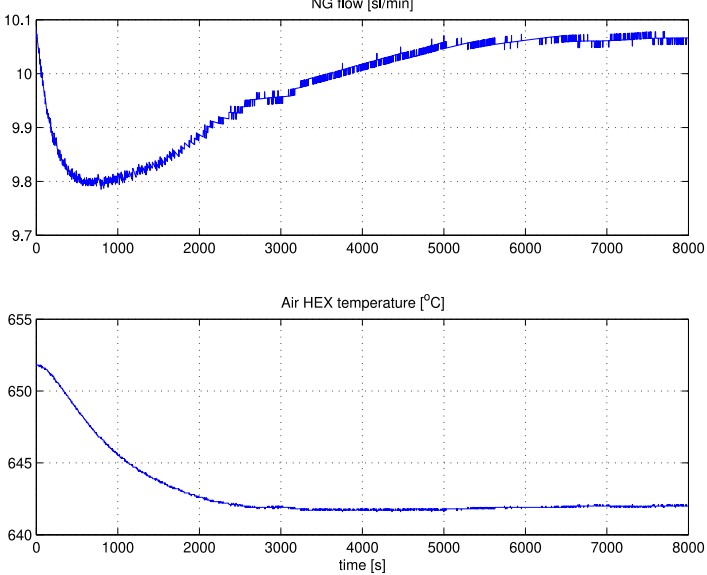

**Figure 27.** Closed-loop response of Air HEX control loop.

## 6. Discussion

Adoption of the MOMI tuning method for integrating processes (IP) was presented in this paper. The closed-loop responses on several different IP models showed that the proposed method resulted in relatively fast closed-loop responses while retaining system stability. Furthermore, the proposed method was also compared to existing tuning methods that are suitable for IP. By using simulated examples, the two distinctive advantages of the proposed method over existing approaches were shown, namely, the closed-loop responses were evaluated by 5% settling time and the ITSE criterion. The comparison with other methods revealed that the proposed method was very good.

Another advantage is that the MOMI method does not require an explicit process model. Besides the calculation of the controller parameters from the general transfer function with pure time-delay, a process open- or closed-loop time response can also be used to calculate the controller parameters. Moreover, both approaches were equivalent, and the latter did not introduce any error in the calculation of controller parameters. Since the computation of the controller parameters is based on relatively simple analytical equations (in the frequency- or time-domain), the proposed tuning method for IP is useful in practice, especially for less-demanding hardware like simpler PLC controllers. Furthermore, reference weighting factor $b$ allows users to emphasise disturbance rejection, reference following, or something in between.

The method was tested on a control system for charge-amplifier drift compensation, laboratory plant, industrial autoclave, and solid-oxide fuel cell. The responses on all processes were relatively fast and highly damped.

In future work, we will concentrate on additionally improving the tracking response while retaining the best disturbance-rejection performance. As already mentioned in Section 2, this can be done by implementing the reference pre-filter instead of reference weighting factor $b$. Although a pre-filter solution (realisation and parameter computation) is much more complex than reference weighting factor $b$, it could be beneficial to obtain the overall optimal response if the process model is well defined.

**Author Contributions:** Conceptualisation, T.K. and D.V.; software, T.K. and D.V.; validation, T.K. and D.V.; formal analysis, T.K. and D.V.; investigation, data curation, and writing, all authors; supervision, D.V. All authors have read and agreed to the published version of the manuscript.

**Funding:** This work was carried out within research programs P2-0001 and PR-07603, financed by the Slovenian Research Agency.

**Acknowledgments:** The authors would like to thank Đani Juričić for the helpful discussions.

**Conflicts of Interest:** The authors declare no conflict of interest.

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
