# Peer review of "Parametric and Nonparametric PI Controller Tuning Method for Integrating Processes Based on Magnitude Optimum"

_applsci, doi:10.3390/app10041443_

Round 1

Reviewer 1 Report

Aimed at both reference tracking and disturbance rejection, the proposed method deals with integrator systems with time delay. Not only results are compared with other methods, but also are validated theoretically and experimentally. The proposed method applies to a lot of industrial processes.

The quality of this paper is indeed rarely superb. My additional comments that may be included for authors are as follows:

In the line next to Equation (63), TF should be italics written in order to be consistent with that in Equation (63). In addition, "supervisory control" in the title of Figure 27 should be explained.

Reviewer 2 Report

The authors presented Magnitude Optimum Multiple Integration (MOMI) tuning method for integrating processes using 2-DOF controller. The basic idea is to maintain the closed-loop magnitude as flat as possible over a wide frequency band. The general stable transfer function with first-order time delay is developed in infinite Taylor series and represented by A_i the so-called "characteristic areas". The authors propose using 2-DOF controller with a reference weighting factor b. When b=0, the controller does not introduce additional zeros into closed-loop transfer function, while when b=1, the controller corresponds to the classical PI controller. The proposed method is compared to some other methods in the literature and experimental results for automated drift compensation of the charge amplifier, hydraulic laboratory plant and an industrial autoclave are included. The paper is well written.

I have the following comments:

- The scientific contribution of the paper is only introducing a reference weighting factor b. The magnitude optimum can be applied to general transfer function with any rational transfer function as a controller. 

- When selecting b=0, the magnitude optimum has better disturbance rejection properties (faster response) since the controller zero is not put close to the dominant poles of the process. When b = 1, usually controller zero is put close to the dominant poles of the process, increasing the speed of response to the reference change. However, disturbance rejection properties are worse (the nominator of the transfer function for disturbance rejection does not have the zeros at the same place as the transfer function for reference tracking)

-  I believe that a better performance could be obtained by selecting the controller with (b = 0), setting the controller gains by the magnitude optimum and including an additional n-th order compensator where the parameters of the compensator are tuned using the magnitude optimum on the transfer function computed as a product of the compensator and the closed loop transfer function. In that way a faster response to the reference change would be obtained keeping the best obtained disturbance rejection properties. 

- u_0, y in (17) is undefined
- grid is missing on some figures.

Reviewer 3 Report

Nowadays it is difficult to imagine any industrial process without automation. Automated control is important to ensure that the process is carried out properly, while ensuring quality. The authors of the reviewed paper have presented a controller tuning method called Magnitude Optimum Multiple Integration (MOMI), primarily focusing on integrating systems.

The authors have correctly defined the purpose of the presentation and consistently achieved it. The references consist of 67 items and cover a large period of time. The content of the paper is interesting, the issues are explained in detail and the considerations are supported by examples - both theoretical and experimental. In the discussion the authors indicate the strengths of the method used, but - comparing it with other tuning methods - they also formulate critical conclusions. Due to the comprehensive and (in my opinion) thorough discussion of the method's foundations as well as the results of research, the paper may be interesting for a wide audience.
Looking at the reviewed paper from a critical point of view, one can say that it describes a successful attempt to implement a certain method rather than its development. Therefore, more emphasis should be placed on the part describing experiments on real objects (Section 5). In this case, the description of the research methodology and the configuration of the test bench is missing - for example, the way of obtaining the controller parameters is sufficiently described, but it is not clear how the controller was implemented and how the output data were collected.

The visual side of the reviewed paper should be evaluated positively, but I have noticed a few minor details that require improvement:

* it looks better when the page starts with text, not formula,
* on the page 7, the caption of the drawing passed to the next page,
* try to avoid large, empty areas in the document,
* graphs: the  dash-dash lines and dash-dot-dash lines do not differ significantly from each other - especially in black-and-white printing there is a problem with distinguishing them.
